# Visualizing hot carrier dynamics by nonlinear optical spectroscopy at the atomic length scale

Yang Luo [1,7], Shaoxiang Sheng [1,7], Andrea Schirato [2,3,7], Alberto Martin-Jimenez[1,4], Giuseppe Della Valle [2,5] ✉, Giulio Cerullo [2,5], Klaus Kern [1,6] & Manish Garg [1] ✉

Probing and manipulating the spatiotemporal dynamics of hot carriers in nanoscale metals is crucial to a plethora of applications ranging from nonlinear nanophotonics to single-molecule photochemistry. The direct investigation of these highly non-equilibrium carriers requires the experimental capability of high energy-resolution (~ meV) broadband femtosecond spectroscopy. When considering the ultimate limits of atomic-scale structures, this capability has remained out of reach until date. Using a two-color femtosecond pump-probe spectroscopy, we present here the real-time tracking of hot carrier dynamics in a well-defined plasmonic picocavity, formed in the tunnel junction of a scanning tunneling microscope (STM). The excitation of hot carriers in the picocavity enables ultrafast all-optical control over the broadband (~ eV) anti-Stokes electronic resonance Raman scattering (ERRS) and the four-wave mixing (FWM) signals generated at the atomic length scale. By mapping the ERRS and FWM signals from a single graphene nanoribbon (GNR), we demonstrate that both signals are more efficiently generated along the edges of the GNR − a manifestation of atomic-scale nonlinear optical microscopy.

Nonradiative decay of optically excited plasmons in nanoscale metals can produce hot carriers – highly energetic electrons and holes whose energy distribution deviates significantly from the equilibrium distribution[1–5]. Efficiently harvesting the energy of these hot carriers is the key to a broad range of emerging applications, e.g., photocatalysis[6,7], photovoltaics[8], and below-bandgap photodetection[9]. Moreover, photogenerated hot carriers can be exploited to modulate nonlinear optical processes in plasmonic[10] or semiconductor[11] nanostructures by all-optical means and at an ultrahigh speed (~ 10 THz), a key functionality for the next generation of nanophotonic devices based on active metamaterials[12].

Albeit the exploitation of hot carriers from nanoscale systems has tremendous potential, optimizing and realizing nanostructures tailored for the above-mentioned applications continues to be challenging. The primary rationale behind this roadblock is the inability to directly investigate hot carriers at their intrinsic spatial length (~ Å), time (~ fs) and energy (~ eV) scales[13,14], due to the difficulty in achieving all three resolutions simultaneously in state-of-the-art experiments.

Recent advances in combining ultrashort laser pulses with scanning probe microscopy techniques have enabled attaining femtosecond temporal and atomic-scale spatial resolutions, simultaneously[15–24]. However, probing the hot carrier dynamics also necessitates achieving

[1]Max Planck Institute for Solid State Research, Stuttgart, Germany. [2]Dipartimento di Fisica, Politecnico di Milano, Milano, Italy. [3]Department of Physics and Astronomy, Rice University, Texas, USA. [4]Instituto Madrileño de Estudios Avanzados en Nanociencia (IMDEA Nanociencia), Madrid, Spain. [5]Istituto di Fotonica e Nanotecnologie – Consiglio Nazionale delle Ricerche, Milano, Italy. [6]Institut de Physique, Ecole Polytechnique Fédérale de Lausanne, Lausanne, Switzerland. [7]These authors contributed equally: Yang Luo, Shaoxiang Sheng, Andrea Schirato. ✉e-mail: giuseppe.dellavalle@polimi.it; mgarg@fkf.mpg.de

this capability over a broad spectral range[4] (eV) with high energy resolution (meV)[25,26]. The broad spectral range and high energy resolution are crucial as hot carriers exhibit a complex time-dependent behavior across a wide range of energies, and capturing the subtleties of this spectro-temporal evolution is critical to understanding and manipulating their dynamics.

Here, we introduce atomic-scale broadband femtosecond nonlinear spectroscopy and use it to directly probe the hot carrier dynamics at the atomic length and femtosecond time scales in the plasmonic picocavity[27] of a scanning tunneling microscope (STM). Hot carriers generated by the nonradiative decay of photoexcited localized surface plasmons (LSPs) were characterized via the emitted anti-Stokes spectrum over a broad spectral range (~ eV) with ~1 meV energy resolution. In order to dynamically control this emission from the picocavity, we performed a two-color pump-probe experiment, where the pump laser pulse controls the hot carrier density, and the spectrally separated probe pulse gives rise to anti-Stokes electronic resonance Raman scattering (ERRS) starting from the out-of-equilibrium carrier distribution induced by the pump pulse. Our analysis reveals that the dominant contribution to the ultrafast modulation of anti-Stokes emission arises from photoexcited hot carriers exhibiting a nonthermal distribution, with energies significantly far from the Fermi level. The unique design of our two-color experiment allows the concurrent detection of atomically localized four-wave mixing (FWM) signals, enabling a precise temporal clocking of the hot carrier dynamics.

We performed atomic-scale microscopy of the anti-Stokes and FWM signal intensities in a single graphene nanoribbon (GNR)[28]. The anti-Stokes as well as FWM signals were dramatically enhanced at the edges of the GNR compared to its interior, which can be attributed to the higher local density of states (DOS) at the edges[28–30]. The enhanced FWM signals at the GNR edges also connote to the fact that higher local DOS leads to a higher nonlinear susceptibility ($\chi^{(3)}$), which varies at the atomic scale[31]. This observation opens new avenues for atomic-scale nonlinear optics, and for the disclosure of a new topic: ultrafast nonlinear picophotonics, affording unique opportunities in a variety of contexts, from the direct investigation of non-equilibrium light-matter interactions in complex quantum materials, to the development of robust strategies for hot carriers harvesting in single molecules and the next generation of active metasurfaces with deep-sub-wavelength meta-atoms.

## Results

### Nonlinear photonics in a plasmonic picocavity
In our experiments, ultrashort laser pulses (wavelength range, $\lambda$ ~ 715–725 nm, pulse duration, $\tau$ ~ 80 fs) illuminate a plasmonic tunnel junction formed between a Au nanotip and a Au(111) surface of an STM, as schematically shown in Fig. 1a (see Supplementary Fig. 1 and Supplementary Note 1 for further details). The interaction with the ultrashort laser pulses generates non-equilibrium hot carriers in the plasmonic tunnel junction, whose energy distribution deviates significantly from the equilibrium Fermi-Dirac distribution. These hot carriers evolve on the femtosecond timescale and are spatially localized at the tunnel junction, as shown by the evaluated photo-absorption pattern (inset of Fig. 1a, see Supplementary Note 2, 3 for further details). The inelastic scattering of the hot carriers with the incident laser pulse leads to the generation of photons whose energies are higher than that of the exciting laser pulses (Fig. 1b), producing an anti-Stokes signal.

The anti-Stokes spectrum measured from the picocavity for exciting laser pulses of ~ 37 pJ energy is shown by the blue curve in Fig. 1c. The LSP resonance spectrum measured via electroluminescence by applying a bias of 3 V to the plasmonic picocavity, in the absence of the exciting laser pulse, is shown by the red curve in Fig. 1c. To enable a direct comparison between the two spectra, we

intentionally kept the bandwidth of our laser pulses narrower (~ 25 meV) in this measurement, so that the spectrum of the laser pulses did not influence the spectral distribution of the generated hot carriers. We note that the bias voltage applied to the STM junction was kept low (~ 100 mV) when measuring the anti-Stokes spectra, to avoid electroluminescence in the interested spectral range[32–35]. The simulated local photonic density of states (LPDOS) of the plasmonic picocavity, shown by the green curve in Fig. 1c (see Supplementary Note 2 for the details of the calculations), exhibits a reasonable agreement with the measured spectra of anti-Stokes signal[36] as well as of the LSP resonance[37]. It is worth mentioning that fine structures present in the anti-Stokes spectrum and the LSP in Fig. 1c plausibly originate from the atomic-scale structural variations at the very end of the apex of the nanotip, which were not considered in the simulations (green curve in Fig. 1c).

The spatial confinement of the hot carriers in the picocavity was investigated by controllably increasing the size of the cavity. The variation of the spectral intensity of the anti-Stokes signal measured as a function of the increasing tip height ($\Delta z$) is shown in Fig. 1d. The spectral intensity almost vanishes on a relative increase of the cavity size by ~ 4 Å. The variation of the anti-Stokes signal can be fitted with an exponential function, $I_{aS} \propto \exp(-k \cdot \Delta z)$, yielding a decay constant of $k$ ~ 0.78 Å$^{-1}$. This observation clearly points out that the hot carriers responsible for the anti-Stokes signal are confined within the localized plasmonic hotspot between the nanotip and the Au(111) surface (see also Supplementary Note 3 and Supplementary Fig. 10).

There are three plausible underlying mechanisms which could lead to anti-Stokes signal from hot carriers: (i) multiphoton absorption ($n \geq 2$) from the exciting laser pulses, followed by interband recombination of the carriers in the conduction and valence bands of Au, giving rise to anti-Stokes emission[38,39]; (ii) intraband recombination of the hot carriers in the Au conduction band[40], which can be treated as a two-dimensional hot electron gas[41]; and lastly (iii) electronic resonance Raman scattering from the hot carriers[38].

To elucidate the physical mechanism underlying the anti-Stokes signal from the hot carriers, we performed an excitation fluence dependence experiment, where the anti-Stokes spectra were measured as a function of the increasing energy of the exciting laser pulses. The spectrally integrated intensity of the anti-Stokes signal exhibits a quadratic dependence on the laser pulse energy, as shown in Fig. 1e.

The plasmonic response of the cavity, as shown in Fig. 1c indicates that the interband transitions (threshold ~ 500 nm)[42,43] between the conduction ($sp$-band) and the valence band ($d$-band) are unlikely, as they would fall in a different spectral range. In the case of intraband recombination, the power-law exponent in the power-scaling experiment should vary linearly with the energy of the photons in the anti-Stokes spectra[41]. However, Supplementary Fig. 2 in the Supplementary Information shows negligible variations in the power-law exponent with respect to the measured photon energies, indicating that intraband recombination of the hot carriers is not the mechanism responsible for the anti-Stokes signal. The third mechanism, electronic Raman scattering, is consistent with our observations. In this scenario, the spectral intensity of the anti-Stokes signal is proportional to both the population of hot carriers and the power of the incident laser[38,44]. Since the population of hot carriers is also proportional to the incident laser intensity, eventually, the spectral intensity of the anti-Stokes signal exhibits a quadratic dependence on the laser power. The electronic states available to hot electrons and holes in Au form a continuum, thus, the electronic Raman scattering occurs under resonance conditions, making 'electronic resonance Raman scattering' (ERRS) the precise description of the physical mechanism behind the anti-Stokes signal observed in this study. This interpretation is further corroborated by the model we developed to describe the measured signals (comprehensive details can be found in the Supplementary Information, Supplementary Note 4), suggesting that the anti-Stokes signal

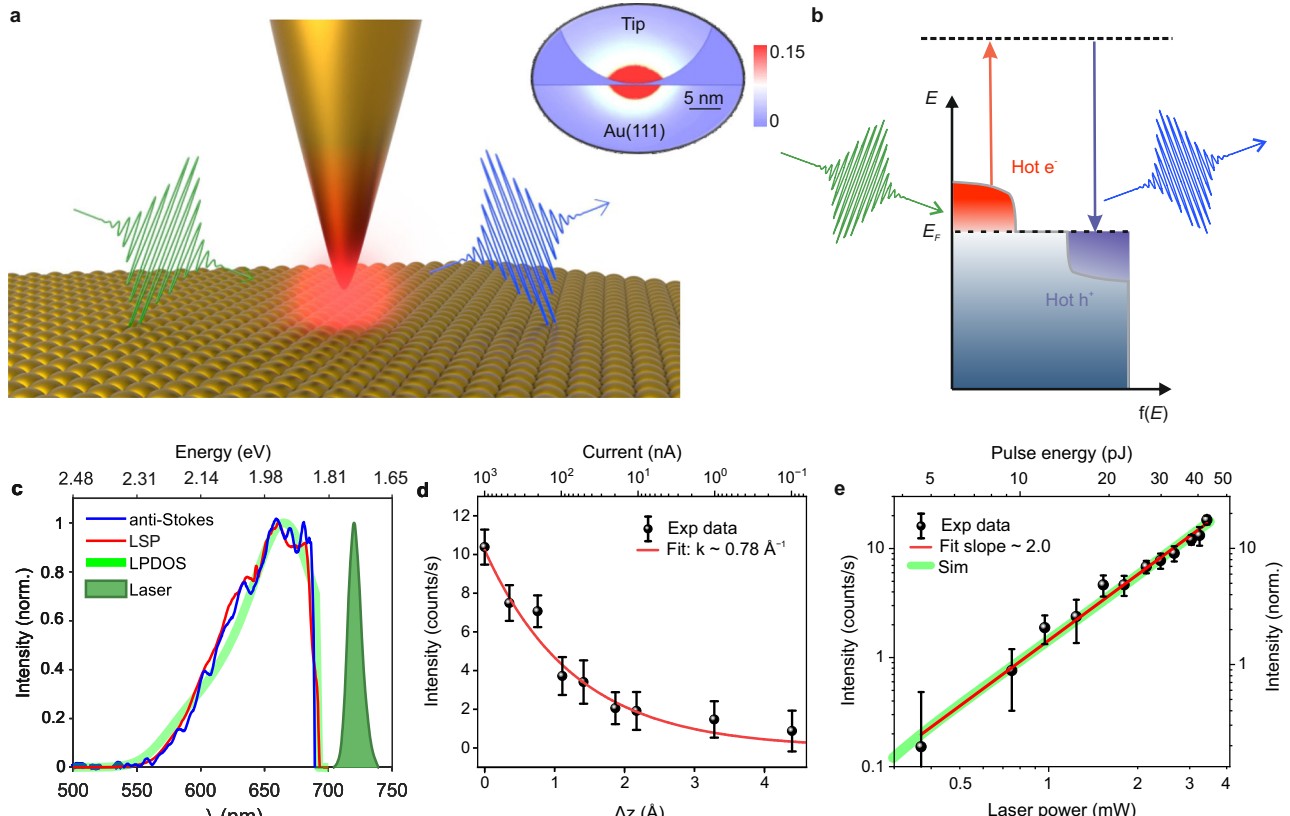

**Fig. 1 | Ultrafast hot-carrier driving of nonlinear optical processes in a plasmonic picocavity. a** Schematic depiction of the hot carrier photogeneration in the plasmonic tunnel junction of a scanning tunneling microscope (STM) by ultrashort laser pulses (green), ensued by an anti-Stokes electronic Raman scattering (blue). Inset: Simulated spatial distribution of the electromagnetic dissipated power across the picocavity (normalized), showing the extreme localization of the optical modes enabled by the plasmonic picocavity. **b** Schematic illustration of hot carriers generation and anti-Stokes light emission: energy distribution of the non-equilibrium hot carriers (electrons and holes) under ultrashort laser pulse excitation, leading to anti-Stokes light emission. $E_F$: Fermi energy level. **c** Comparison of the anti-Stokes spectrum (blue curve) from the plasmonic junction with the local surface plasmonic resonance (red curve), and the calculated local photonic density of states (LPDOS) of the plasmonic picocavity (light green curve). The spectrum of the exciting laser pulses is shown by the filled green curve. The top *x*-axis represents the

spectral axis in eV. **d** Variation of the measured anti-Stokes signal intensity at ~ 680 nm (black dots) as a function of the increasing plasmonic picocavity size (Au nanotip – Au(111) sample distance) and with the corresponding decreasing tunneling current (top *x*-axis). An exponential fit of the measured anti-Stokes signal (red curve) yields a decay constant of ~ 0.78 Å⁻¹. Δz = 0 Å in the plot represents the height of the nanotip from Au(111) surface at the tunneling condition of 1 μA at 100 mV. **e** Variation of the measured (black dots) and simulated (light green curve) anti-Stokes signal intensity as a function of increasing incident laser power, plotted in a dual-logarithmic scale, respectively, and the corresponding quadratic fit of the measured data (red curve). The STM junction was operated under constant current mode with the set tunneling current of 500 nA at 100 mV. Parameters of the exciting ultrashort laser pulse: wavelength range: λ - 715–725 nm, pulse duration ~ 80 fs, power ~ 3 mW. Error bars in (**d**) and (**e**) represent the standard deviation from the integrated spectral area.

from the picocavity can indeed be rationalized by the ERRS mechanism. The simulated power-scaling of the anti-Stokes intensity upon single-pulse excitation (green curve in Fig. 1e) precisely follows the experimentally measured quadratic dependence (black dots and red curve in Fig. 1e).

## Broadband femtosecond nonlinear optical spectroscopy at the atomic scale

To time resolve the relaxation dynamics of hot carriers in the picocavity, we performed a broadband two-color pump-probe experiment, as schematically shown in Fig. 2a. Laser pulses with the spectral range of ~ 830–870 nm (~ 30 fs), hereafter, referred to as pump pulse, were used to photoexcite the hot carriers; whereas laser pulses with the spectral range of ~ 715–750 nm (~ 30 fs), hereafter, referred to as probe pulse, were used to track the time evolution of the hot carriers by ERRS (see "Methods"). The spectral gap between the pump and probe pulses ensures that the pump pulse effectively excites the hot carriers without intervening with the measured anti-Stokes signal, which is exclusively triggered by the probe pulse. Moreover, the extreme localization of the electromagnetic fields even far from its LSP (see Supplementary Fig. 10) in

the picocavity ensures that both the pump and the probe pulses, although non-resonant, effectively interact with the system.

The time evolution of the non-equilibrium distribution of the hot carriers as a function of the delay between the pump and probe pulses is pictorially depicted in Fig. 2a. At negative time delays (τ < 0), when the probe pulse precedes the pump pulse, there is barely any change in the distribution of the hot carriers. In contrast, at zero and positive time delays (τ ≥ 0), the probe pulse enables probing the density and the energy distribution of the hot carriers excited by the pump pulse through the time-evolving anti-Stokes spectra. In addition, at zero delay, nonlinear optical processes (three-wave mixing and four-wave mixing (FWM)) can occur between the pump and probe pulses in the picocavity. In the sketch of Fig. 2a, we specifically illustrate the FWM signal (blue arrow) produced by two interactions with the probe pulse (upward transition, green arrows) and one interaction with the pump pulse (downward transition, orange arrow), as this is the only wave-mixing process that can be observed within the spectral range of interest, given the bandwidths of the pump and probe pulses, as shown in Fig. 2b.

The contributions of the FWM (~ 630–645 nm spectral region) and the hot carrier signal (~ 660–690 nm spectral region) in the anti-

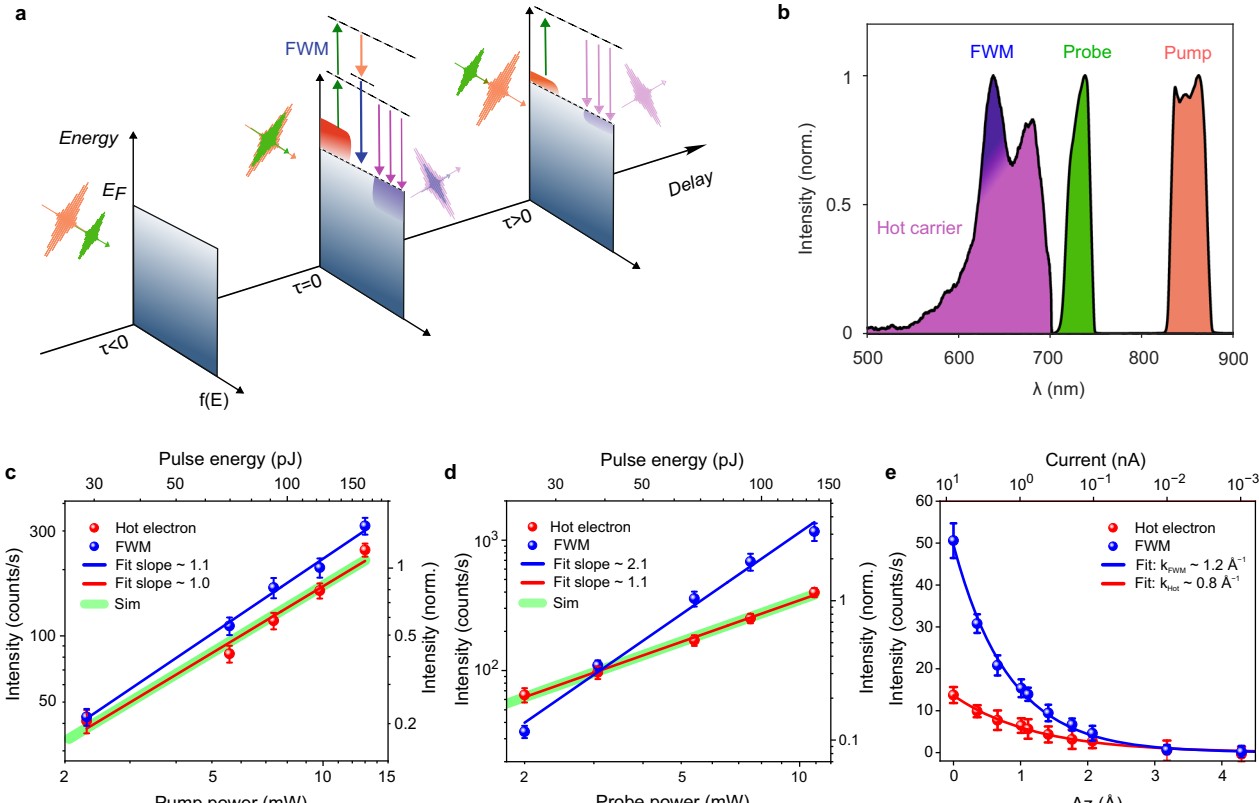

**Fig. 2 | Two-color broadband nonlinear spectroscopy of hot carriers. a** Sketch of the temporal evolution of the energy distribution of the hot carriers in the pico-cavity as a function of the pump-probe delay. Left panel: electron distribution when the probe pulse (green curve) precedes ($\tau < 0$) the pump pulse (orange curve). Middle-panel: hot carrier generation on excitation by the pump pulse, followed by anti-Stokes emission (vertical purple curves) upon interaction of the hot carriers with the probe pulse (vertical green curve, $\tau = 0$). Four-wave mixing (FWM) process (vertical blue curve) occurs when the pump (orange) and probe pulses (green) temporally overlap. Right panel: relaxation of the hot carriers after the pump pulse excitation ($\tau > 0$), which is tracked in real-time by the evolving anti-Stokes signal generated by the probe pulses. **b** Spectra of the pump pulse (orange curve), probe pulse (green curve), and the anti-Stokes spectrum measured at the temporal overlap (zero delay) between the pump and the probe pulses. The purple and blue shaded regions in the anti-Stokes spectrum depict the contributions of the hot carriers and the FWM signals, respectively. **c**, **d** Variation in the intensity of the hot carrier (red dots) and FWM (blue dots) contributions in the anti-Stokes spectra as a function of the increasing fluence of the pump (**c**) and probe (**d**) pulses, plotted in a dual-logarithmic scale, respectively. Green curves show the numerically calculated variation in the intensity of the hot carrier contribution (at 680 nm) in the simulated anti-Stokes spectra upon change of the fluence of the pulses. **e** Variation in the intensity of the hot carrier (red dots) and FWM (blue dots) contributions in the anti-Stokes spectra as a function of increasing tip height ($\Delta z$), with the corresponding decreasing tunneling current (top $x$-axis). $\Delta z = 0$ Å represents the height of the nanotip from the Au(111) surface at the tunneling condition of 8 nA at 100 mV. Probe and pump laser powers were fixed at 3.1 mW and 7.35 mW, respectively, in (**c**, **d**, and **e**). The delay between pump and probe pulses was set to be zero fs in (**c**, **d**, and **e**). Error bars in (**c**, **d**, and **e**) represent the standard deviation from the integrated spectral region.

Stokes spectra were investigated by individually varying the fluence of the pump and probe pulses at zero time delay between them. When increasing the fluence of the pump pulse, while keeping the fluence of the probe pulse fixed, both the FWM and hot carrier contributions exhibit a linear dependence, as shown in Fig. 2c. Since the pump pulse is responsible for exciting the hot carriers in the picocavity, thus, its linear dependence on the hot carrier contribution is justifiable, as also predicted by our simulation (green curve in Fig. 2c). Similarly, the FWM signal, which involves one interaction with the pump pulse, i.e., stimulating a downward transition ensuing two interactions with the probe pulse, also exhibits a linear dependence on the pump fluence. The FWM signal intensity can be expressed as: $I_{FWM} \propto |\chi^{(3)} E_{probe} E_{probe} E^*_{pump}|^2$, where $E_{probe}$ and $E_{pump}$ are the electric fields of the probe and pump pulses, respectively. $\chi^{(3)}$ is the third-order nonlinear susceptibility.

In contrast, when varying the power of the probe pulses ($P_{probe}$), the spectral intensity of the FWM contribution ($I_{FWM}$) in the anti-Stokes spectra varies quadratically ($I_{FWM} \propto P^{\alpha}_{probe}$, $\alpha \sim 2.1$, blue curve in Fig. 2d), which is consistent with the two interactions with the electric field of the probe pulses in the measured FWM process. However, the spectral intensity of the hot carrier contribution in the anti-Stokes spectra shows a near-linear dependence ($\alpha \sim 1.1$) on the fluence of the probe pulses (red curve in Fig. 2d), differing from the quadratic dependence observed in the single-pulse power-scaling experiment shown in Fig. 1e.

The fluence dependence measurements further substantiate our interpretation of the electronic resonance Raman scattering as the mechanism behind the anti-Stokes signal. Here, the role of the probe pulse is just to perform ERRS from the hot carriers pre-excited by the pump pulse. In the single-pulse experiment (Fig. 1e), hot carriers are both generated and undergo ERRS by the same pulse, thus leading to a quadratic power scaling. However, in the two-color pump-probe experiment, the hot carriers are pre-generated by the pump pulse, resulting in a much less nonlinear dependence ($\alpha \sim 1.1$) in the power-scaling experiment with the probe pulses, as also confirmed by the simulations, green curve in Fig. 2d.

The spatial extent of the localization of the FWM and hot carrier signals in the anti-Stokes spectra at zero time delay between the pump and probe pulses was explored by controllably increasing the tip-sample distance. Figure 2e shows the variation in the spectrally integrated intensity of both the FWM and hot carrier signals as a function of increasing tip height, achieved by decreasing the tunneling current

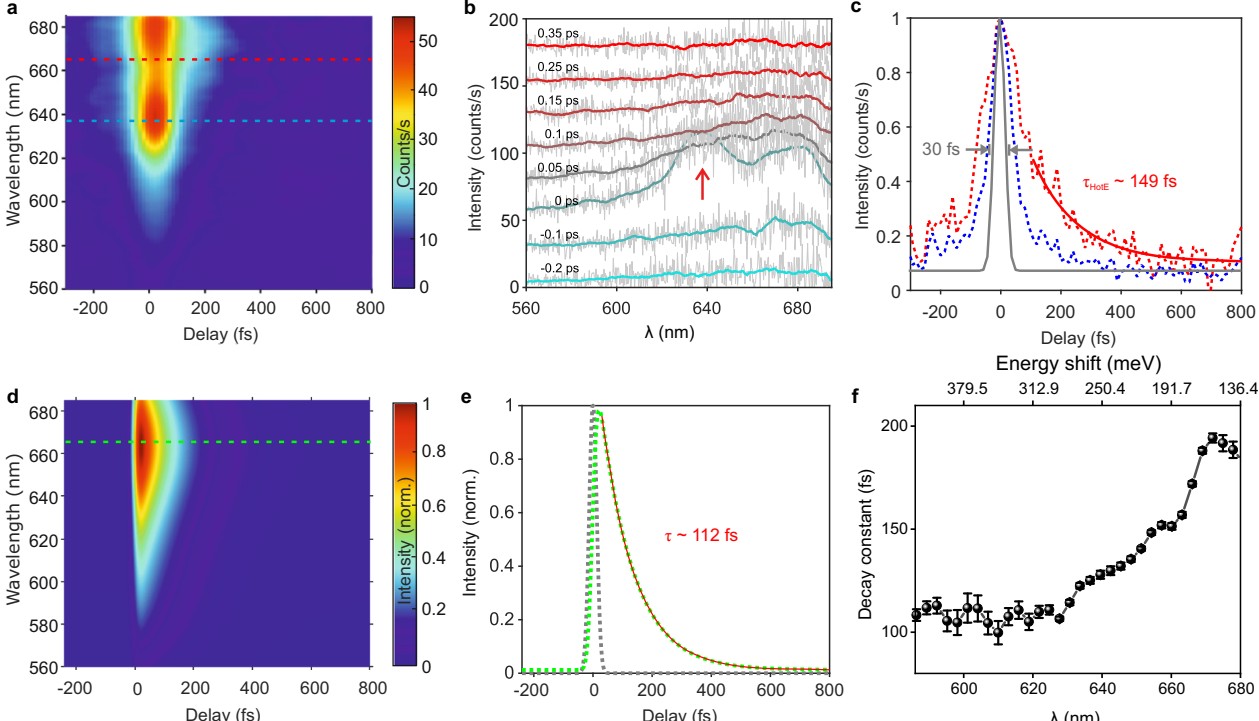

**Fig. 3 | Atomic-scale tracking of hot carrier dynamics. a** A series of anti-Stokes spectra measured as a function of the delay between the pump and the probe pulses. Pump pulse parameters: λ ~ 830–870 nm, pulse duration ~ 30 fs, power ~ 5.8 mW (72 pJ). Probe pulse parameters: λ ~ 715–750 nm, pulse duration ~ 30 fs, power ~ 2.6 mW (32 pJ). STM was operated in the constant current mode with a tunneling current of 8 nA and a bias of 100 mV. **b** Representative anti-Stokes spectra from the measurement shown in (**a**) at different delays between pump and probe pulses, as annotated on top of each spectrum. **c** Spectral intensity variation of the anti-Stokes spectra at ~ 665 nm (hot carrier, dashed red curve) and at ~ 640 nm (FWM, dashed blue curve) as a function of the delay between pump and probe pulses, indicated by horizontal dashed red and blue lines in (**a**) respectively. The gray curve represents the duration of the probe pulse (see "Methods"). An

exponential fit (solid red curve) of the temporal cross-cut reveals a relaxation time of ~ 149 fs. **d** Simulated anti-Stokes spectra as a function of the delay between the pump and the probe pulses. The parameters of the pump and probe pulses used in the simulations are identical to the experimental ones. **e** Temporal cross-cut at 665 nm from the simulation shown in (**d**) (horizontal dashed green curve). A relaxation time of ~ 112 fs is estimated from the exponential fit of the temporal cross-cut (solid red curve). **f** Variation of the measured relaxation times of the anti-Stokes signal as a function of various wavelengths. The top x-axis represents the energy shift of the anti-Stokes signal with respect to the central wavelength (~ 735 nm) of the probe pulse. The error bars indicate the standard deviation from the fit.

in the constant current operation mode of the STM. Both the FWM and hot carrier signals decrease dramatically upon increasing the tip height. The FWM signal decays with a constant of ~ 1.2 Å$^{-1}$ (blue curve in Fig. 2e), which is swifter than the decay rate of the hot carrier signal of ~ 0.8 Å$^{-1}$ (red curve in Fig. 2e). The swifter decay of the FWM signal suggests higher sensitivity to the changes in the size of the picocavity in contrast to the hot carrier contribution. This is because FWM depends on the fourth- and second-powers of the locally enhanced probe and pump electric fields, $I_{FWM} \propto |\chi^{(3)} E_{probe} E_{probe} E_{pump}^*|^2$, while the hot carrier contribution (ERRS signal) depends quadratically on the electric fields of both the pump and the probe pulses.

## Energy dependent relaxation dynamics of the hot carriers

A series of anti-Stokes spectra measured as a function of the delay between the pump and probe laser pulses is shown in Fig. 3a. Figure 3b shows anti-Stokes spectra at several representative delays between the pump and probe pulses. The spectral feature at ~ 640 nm observed at the zero delay between the pump and probe pulses, also indicated by the red arrow in Fig. 3b, results from the FWM process between the two pulses. This nonlinear FWM signal generated by the pump and the probe pulses facilitates the determination of the absolute time delay between the pulses in the experiment, with the maxima in the position of the FWM signal signifying the absolute time zero in the measurements[45,46]. This enables us to clock the anti-Stokes signal from the probe pulses following the pump-driven excitation of the hot carriers.

A comparison between the temporal cross-cuts of the hot carrier signal at ~ 665 nm and the FWM signal from the spectral region at ~ 640 nm, obtained from the pump-probe measurement, is shown in Fig. 3c. The probe pulses lead to anti-Stokes scattering from the hot carriers generated by the pump pulses almost instantaneously, limited only by the time-resolution of the experiment, which is ~ 30 fs, as indicated by the gray curve in Fig. 3c (see "Methods"). An exponential fitting of the decay profiles of the measured anti-Stokes spectra (thick red curve in Fig. 3c) reveals the relaxation time of the hot carriers responsible for the anti-Stokes signal at ~ 665 nm to be approximately 149 fs.

Figure 3d shows the simulated time-resolved anti-Stokes spectra, that we modeled by combining a rate-equation-like description of the ultrafast dynamics of hot electrons with a formulation of the ERRS cross-section of the picocavity (see Supplementary Note 4, 5 and 8 for the details). For the hot carrier dynamics, we built upon the well-established Three-Temperature Model[47], and extended it to a spatio-temporal model to account for the peculiar ultrafast spatial diffusion of carriers in the picocavity region. For the ERRS, we adapted approaches previously reported[36,48,49] to assess the broadband anti-Stokes Raman scattering. The details of our modeling approach are provided in the Supplementary Information. Our dynamical simulations exhibit a reasonable agreement with the measured anti-Stokes ERRS signal, both in terms of the spectral shape (besides the FWM fingerprint, not included in our model) and the temporal evolution. In particular, a temporal cross-cut at ~ 665 nm from the simulation is shown in Fig. 3e. Fitting the exponential decay profile reveals a

thermalization time of the hot carriers to be ~ 112 fs, which matches reasonably with the measured thermalization time of ~ 149 fs. Our model indicates that the observed signal is dominated by the contribution arising from the nonthermalized hot carriers (see Supplementary Note 7 for discussions on the different contributions in the ultrafast modulation of the anti-Stokes emission), while the one from thermalized hot carriers (i.e., closer to the Fermi level) following electron-electron scattering is drastically reduced, due to their spatial diffusion away from the picocavity active region (see Supplementary Note 6 for details). A maximum increase in the electronic temperature of mere ~ 30 K is predicted, following a peculiar spatiotemporal dynamics in the picocavity, effectively washing out the contribution of the thermalized hot carriers in the measured anti-Stokes signal. It is worth mentioning that since the temporal cross-cut in the FWM region (~ 630–645 nm spectral region) also has an underlying contribution from the hot carriers, hence, it also contains information about their relaxation times, as evident from the relatively broad signal at ~ 640 nm (blue curve in Fig. 3c) compared to the accessible time resolution in the measurements (gray curve, see also "Methods"). Isolating the FWM contribution in the anti-Stokes spectra is possible by systematically introducing linear positive dispersion in the pump and the probe pulses (Supplementary Fig. 7 in the Supplementary Information).

As noticeable from the pump-probe measurement shown in Fig. 3a, the relaxation time of the hot carriers are energy dependent. A spectrally resolved exponential fitting of the temporal cross-cuts of the anti-Stokes spectra reveals the energy dependent relaxation times of the carriers, as shown in Fig. 3f. Hot carriers with an energy of ~ 380 meV (~ 600 nm) thermalize with a relaxation time of ~ 100 fs, while those with an energy of ~ 150 meV (~ 675 nm) exhibit a much longer relaxation time of ~ 200 fs. High-energy hot carriers relax much faster compared to the low-energy hot carriers due to the greater availability of electron-electron scattering channels at higher energies[50].

### Atomic-scale microscopy of hot carrier distribution in a single GNR

Lastly, we map the hot carrier and FWM intensity distributions in a single seven-atom-wide graphene nanoribbon (7-AGNR) on the Au(111) surface. We chose GNRs for the study, due to their potential in the development of the next generation of molecular nanoelectronics, where it would be crucial to understand and control hot carrier dynamics at the atomic length scales.

Figure 4a shows the constant current STM topography of a single GNR. The simultaneously measured spatial variation of the spectral intensity of the hot carrier and the FWM signals in the anti-Stokes spectra are shown in Fig. 4b, c, respectively. The anti-Stokes spectra were measured at zero delay between the pump and the probe pulses. Both hot carrier and FWM signals are dramatically enhanced at the edges of the GNR compared to its interior, as also evident from the line profiles shown in Fig. 4d. This enhancement is consistent with the known higher local electronic DOS at the edges of GNRs[30], as also confirmed by differential conductance measurements (see Supplementary Fig. 3 and Fig. 4 in the Supplementary Information). In addition, simulations were performed (see Supplementary Note 8) to exclude purely photonic (or geometric) effects, i.e., modified LPDOS within the cavity, at the origin of this observation.

A higher DOS would make available a larger pool of carriers which can be photoexcited by the incident pump pulses, leading to a stronger anti-Stokes signal from the hot carriers at the edges of the GNR compared to its interior (Fig. 4b). In close analogy to the hot carrier signal in the anti-Stokes spectra, the higher DOS at the edges of the GNR could enhance the nonlinear susceptibility to the incident light. As a result, the nonlinear response linked to the third-order nonlinear susceptibility ($\chi^{(3)}$) exhibits atomic-scale variations within the GNR. This study attests to the critical role of the local DOS in facilitating efficient

hot carrier generation and in enhancing the nonlinear optical response in single molecules.

Time-resolved dynamics were measured at various places over the GNR. Since the GNR lies flat on the Au(111) surface, it is electronically coupled to it, thus making the dephasing times of the excited plasmons and eventually the hot carriers very similar to those observed on a clean Au(111) surface[18]. However, for molecules which are electronically decoupled from the metallic surface, a different hot carrier dynamics is expected. This dynamics would uniquely reflect the contributions of the various electronic levels of the molecules[51] and will be the focus of a future work.

## Discussion

We have introduced broadband (~ eV) femtosecond nonlinear optical spectroscopy and microscopy at the atomic length scale, and have used it to demonstrate the concept of ultrafast hot-carrier mediated modulation of anti-Stokes electronic resonance Raman scattering in a plasmonic picocavity. Our setup enables the direct tracking of the photogenerated hot carriers and mapping of their spatial distribution in a single molecular entity. By conducting a two-color pump-probe spectroscopic measurement, we have determined the relaxation times of the hot carriers in the picocavity to be energy dependent, with high-energy hot carriers relaxing faster than their low-energy counterparts. The atomic-scale mapping of FWM and hot carrier distributions opens the door to accessing the nonlinear optical properties and their associated dynamics in individual molecules and complex quantum materials, which were previously accessible only in ensemble measurements at macroscopic length scales.

Atomic-scale broadband nonlinear spectroscopy is the key to realizing the long-sought after goal of seeing chemistry in motion in single molecules[52]. Furthermore, the technique is ideally suited to probe transient photo-induced ferromagnetism and superconductivity[53,54] at atomic scales, and develop a new generation of ultrafast devices based on active metamaterials[12,55] with picosized meta-atoms[56].

## Methods

### Sample and tip preparation

All the experiments in the current work were performed in a home-built scanning tunneling microscope (STM) operating in ultra-high vacuum (UHV) conditions (~ 5 × 10^{-11} mbar) and at liquid-helium temperature (~ 11 K). Au(111) surfaces were prepared by repeated cycles of sputtering with 1.0 keV Ar$^+$ ions, followed by thermal annealing at ~ 430 °C. Au tips prepared by electrochemical etching were used in all the experiments for the plasmonic enhancement. To fabricate the graphene nanoribbons, 0.5 monolayer of 10,10′-dibromo-9,9′-bianthryl (DBBA) molecules were sublimated on the clean Au(111) substrate held at room temperature. The 7-armchair graphene nanoribbons (7-AGNRs) were obtained by post-annealing the sample at 200 °C for 10 min, then at 400 °C for another 10 minutes[57].

### Optical setup

The ultrafast laser system used in the current work is a Ti:Sapphire oscillator (Element™ 2, Newport Spectra-Physics) which produces laser pulses of ~ 6 fs duration with a bandwidth spanning from 650 nm to 1050 nm at a repetition rate of ~ 80 MHz. Laser pulses with reduced spectral range were generated by bandpass filtering the broadband laser pulses from the oscillator and compressed using chirped dielectric mirror pairs (see Supplementary Fig. 7 for details). A precise delay stage (N-565, Physik Instrumente) was used to control the delay time between the pump and probe pulses. A biconvex lens (diameter: 50 mm; focusing length: 75 mm) was mounted inside the UHV chamber to focus the laser beams onto the apex of the Au tip. The dispersion accumulated by the laser pulses on passing through several dispersive elements in the setup (biconvex lens and UHV window) was pre-compensated by multiple reflections off a pair of chirped dielectric mirrors. A second harmonic

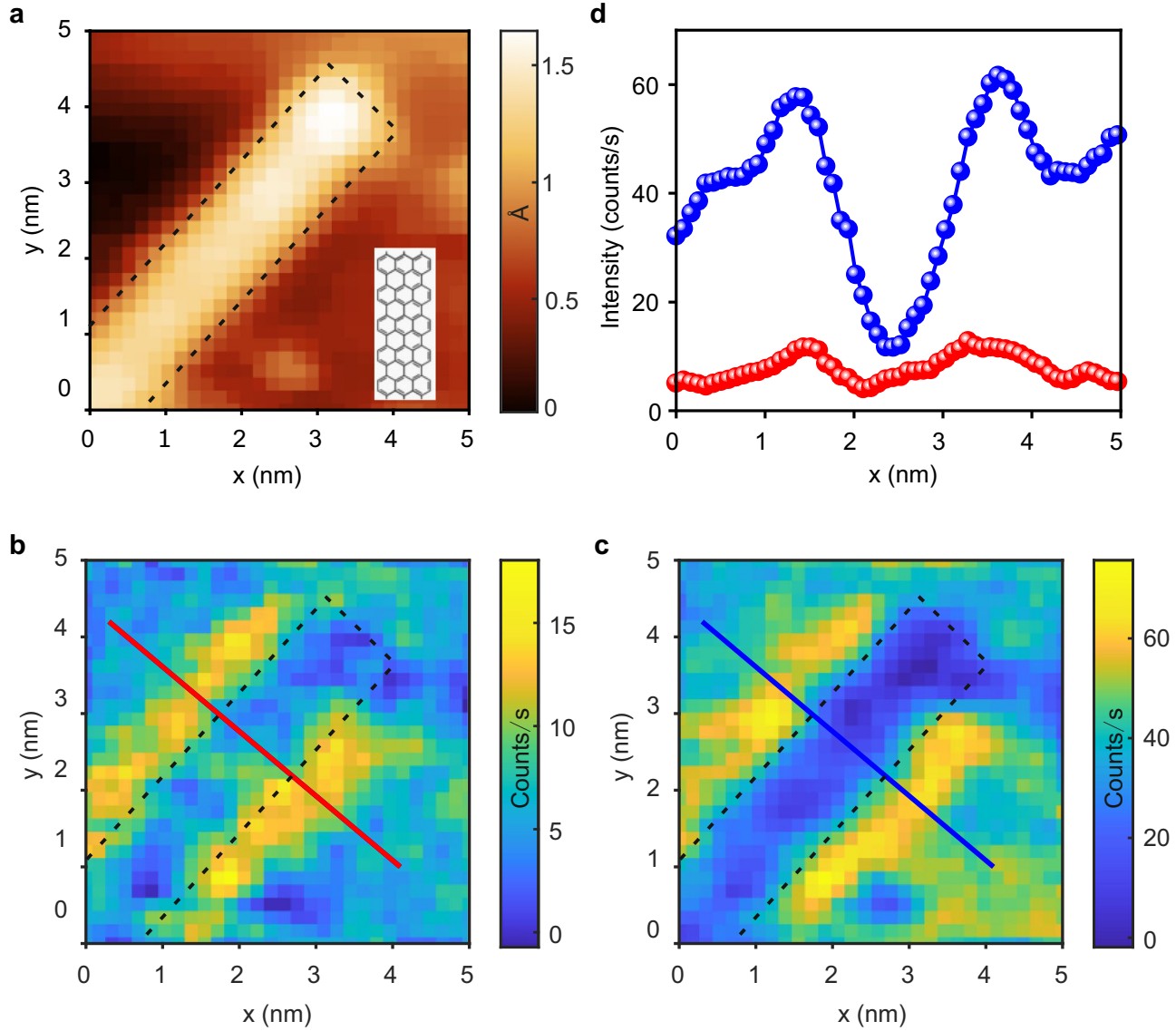

**Fig. 4 | Atomic-scale microscopy of hot carriers and nonlinear susceptibility ($\chi^{(3)}$). a** STM image of a single seven-atom-wide graphene nanoribbon (7-AGNR) on the Au(111) surface. The color bar represents the relative change in the vertical position of the Au nanotip (in Å) while scanning the GNR in the constant current mode of the STM with a tunneling current of 1 nA at 1 V. Bottom-right inset in (**a**) shows the chemical structure of the GNR. **b**, **c** Spatial variation of the intensity of the hot carrier (**b**) and the FWM (**c**) contributions in the anti-Stokes spectra recorded simultaneously with the topography shown in (**a**). The delay between the pump and the probe pulses was set to be zero. The hot carrier and the FWM contributions in the anti-Stokes spectra were evaluated by spectral integration in the wavelength range of 665–675 nm and 635–645 nm, respectively. Dashed black rectangles in (**a**, **b** and **c**) indicate the location of the GNR. **d** Variations of the hot carrier (red) and FWM (blue) signals intensities along the annotated lines in (**b** and **c**), respectively. Probe and pump laser powers were fixed at 2.5 mW and 5.5 mW, respectively.

generation based fringe resolved autocorrelator (FRAC) with an ~20 μm thick BBO crystal was used to measure the duration of the pump and probe pulses with identical dispersion as in the optical path to the STM junction. The duration of the pump and the probe pulses were measured to be ~30 fs (gray curve in Fig. 3c). The anti-Stokes signal was collected through the same biconvex lens and then focused into the entrance slit of a spectrometer (Kymera 328i, ANDOR) and detected by a thermo-electrically cooled charge coupled device (iDus 416, ANDOR). A schematic of the experimental setup is shown in Supplementary Fig. 1 in the Supplementary Information.

### Modeling ultrafast hot carrier dynamics

The non-equilibrium dynamics of the hot carriers was described numerically by an extended Three-Temperature Model[47] (3TM). In general, the 3TM is a semiclassical rate-equation model which details the ultrafast relaxation of plasmonic hot electrons in terms of three internal energetic variables: (i) the excess energy stored in a 'non-thermal' portion of the carrier population, featuring energies as high as the absorbed photon ones; (ii) an increased electronic temperature, associated with an excited Fermi-Dirac occupancy distribution; and (iii) the metal lattice temperature. Here, given the peculiar electro-magnetic mode confinement induced by the picocavity, we adapted the 3TM original formulation to account for the ultrafast spatial diffusion of thermalized hot carriers that additional Finite Element Method (FEM)-based simulations showed to be critical to the electron dynamics. By then, numerically integrating our reduced 3TM and conveniently combining the contributions arising from both the probe and the pump pulses, we retrieved the ultrafast evolution of the nonequilibrium carriers' occupancy distribution. Further details on our model can be found in the Supplementary Information.

## Finite-element electromagnetic simulations

A three-dimensional FEM-based model was developed using commercial software (COMSOL Multiphysics, 6.2) to calculate the optical and electromagnetic properties of the plasmonic picocavity. The system was modeled as a nanoparticle-on-mirror geometry, made of an Au nanosphere (mimicking the STM tip) placed ad a sub-nm distance on top of a flat Au substrate. Both dipolar and plane wave excitations were implemented, enabling us to determine various relevant quantities of the cavity electromagnetic behavior, including the local photonic density of states, the absorption and scattering cross-sections, the photo-absorption spatial patterns and mode volumes. The details of our model are reported in the Supplementary Information.

## Data availability

The data generated in this study are available within the main text and the Supplementary Information.

## Code availability

The details needed to reproduce the computations have been provided in the "Methods" section and the Supplementary Information file.

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

## Acknowledgements

We thank Wolfgang Stiepany and Marko Memmler for technical support. A.S., G.D.V., and G.C. acknowledge financial support by the European Union's NextGenerationEU Program with the I-PHOQS Infrastructure [IR0000016, ID D2B8D520, CUP B53C22001750006] "Integrated infrastructure initiative in Photonic and Quantum Sciences", and from the METAFAST project that received funding from the European Union Horizon 2020 Research and Innovation program under Grant Agreement No. 899673. This work reflects only the author's view, and the European Commission is not responsible for any use that may be made of the information it contains. G.D.V. acknowledges the support from the HOTMETA project under the PRIN 2022 MUR program funded by the European Union – Next Generation EU - "PNRR - M4C2, investimento 1.1 - "Fondo PRIN 2022" - HOT-carrier METasurfaces for Advanced photonics (HOTMETA), contract no. 2022LENW33 - CUP: D53D2300229 0006". A.S. and G.D.V. acknowledge the European Union's Horizon Europe research and innovation program under the Marie Skłodowska-Curie Action PATHWAYS HORIZON-MSCA-2023-PF-GF grant agreement No. 101153856. A.M.J. acknowledges funding from HORIZON-MSCA-2022-PF-01-01 under the Marie Skłodowska-Curie grant agreement No. 101108851.

## Author contributions

Y.L., S.S., A.M.J., K.K., and M.G. built the experimental setup, performed the experiments and analyzed the experimental data. A.S., G.C., and G.D.V. designed and performed the theoretical calculations and analyzed the theoretical data. M.G. conceived the project and designed the experiments. All authors interpreted the results and contributed to the preparation of the manuscript.

## Funding

## Competing interests

The authors declare no competing interests.
