## [Peer Review file · Nature Communications]

Visualizing hot carrier dynamics by nonlinear optical spectroscopy at the atomic length scale

Corresponding Author: Dr Manish Garg

Version 0:

Reviewer comments:

Reviewer #1

(Remarks to the Author)

The authors present a study on light upconversion in plasmonic structures and use it to explore the time-dependent properties of hot electrons excited in gold nanocavities. They use an STM tip coupled to a gold surface to achieve extreme localization of plasmonic fields in picocavities formed within the tunneling gap. Using pump-probe measurements, they characterize the nonequilibrium carriers on ultrafast timescales, shedding light on some of the most hotly debated issues in hot-electron physics such as the timescales of carrier relaxation and the nature of the electronic excitations in these systems. They also perform extensive modeling of the spatiotemporal dynamics of the excited carriers, accounting for various contributions, including both non-thermal and thermal carrier distributions. Finally, they apply this spatiotemporal mapping technique to graphene nanoribbons to image these properties at the atomic scale.

The paper presents an impressive combination of experimental and theoretical work, producing timely and potentially impactful results. However, several substantial issues in the data analysis currently prevent me from recommending the paper for publication.

1. The plasmonic resonance of the system lies around 660–680 nm. This implies that, particularly in the pump–probe experiments, the excitation is non-resonant. Consequently, the discussion of hot-electron effects becomes questionable, because the processes described in Figure 2a appear to be purely plasmonic. Although there is likely some absorption at the pump and probe wavelengths (as shown by Figure S6), it is doubtful that this absorption alone leads to any additional physics beyond what is observed in bulk gold.
2. The slope of 2 observed in Figure S2 can be explained by intraband PL rather than Raman scattering (see Nat Comm , 15, 4468 (2024) or ACS Nano 17, 11439-11453 (2023) for reference). A key question is whether the model presented in the Supporting Information can adequately fit the experimental results shown in Figure S2b.
3. In Figure 1e (green curve), does the simulated slope take into account the hot-electron Raman process and the rate-equation simulations described in the Supporting Information? Clarification on how these factors are incorporated into the modeling would be helpful.
4. In Figure 1, the probe pulse alone is observed to produce anti-Stokes emission. However, in Figure 2, the presence of the pump pulse is required for anti-Stokes emission to be detected. What causes this discrepancy? Is it because of the low power of the probe pulse?
5. What can be inferred from the anti-Stokes scattering transient spectra regarding the carrier distribution—specifically, whether it follows a Fermi–Dirac distribution or a non-Fermi distribution similar to e.g. Nat. Comm., 9, 1853, (2018)? Can the relative population of non-Fermi electrons be estimated as a function of time (for example, from Figure S8c)?
6. The decay constants shown in Figure S8 are presumably determined by the parameters in Equation S10 of the Supporting Information. How are these parameters (e.g. τ_0) chosen or derived from the data?
7. Why does the graphene local density of states (LDOS) influence the electronic processes in gold, given the earlier assertion that anti-Stokes emission originates primarily from hot electrons in Au? Is there additional Raman scattering from

carriers in graphene, or another mechanism at play?

8. Can the lifetimes of electron thermalization as a function of electron energy (shown in Figure 3f) be assessed theoretically? They appear consistent with Fermi liquid theory, so it would be valuable to see theoretical insights on this behavior. Furthermore, the timescale of non-thermal electron decay is notably slow compared with previous literature—does any existing theory explain such slow decay?

9. The results presented in Figure S8 suggest that, under this experimental configuration, the optical response is dominated by a nonthermal, non-Fermi distribution of electrons—an unusual observation compared to most literature reports. These findings should be highlighted more prominently in the main text.

10. Finally, does the tunneling current affect any of the optical or electronic phenomena reported in the paper? A brief discussion or control experiment would help clarify its impact.

Reviewer #2

(Remarks to the Author)

By using nonlinear optical spectroscopy in a plasmonic cavity formed in the tunnel junction of a scanning tunneling microscope (STM), the manuscript by Luo et al. reports a novel method for visualizing hot carrier dynamics at the atomic scale. The authors convincingly demonstrate the ability to track hot carriers and map their spatial distribution, revealing energy-dependent relaxation times and the influence of local density of states. The combination of ultrafast pump-probe spectroscopy with STM offers a unique approach to accessing hot carrier dynamics, with the capability to achieve exceptionally high resolution in both temporal and spatial domains. The experimental results are well supported by theoretical calculations and simulations, providing a comprehensive understanding of the observed phenomena.

I believe that this work represents a significant advance in the field. Therefore, the paper can be considered for publication in Nature Communications if the authors can properly address the following comments or concerns:

(1) In Figure 1c, the experimental LSP electroluminescence and anti-Stokes spectra show some fine structures, while the simulated LPDOS appears smoother. Can the authors comment on the possible origin of this difference?

(2) In the zero-delay experiment presented in Figure 2b, the four-wave mixing (FWM) and anti-Stokes signals exhibit comparable magnitudes despite their distinct physical origins. I wonder if the authors could provide some insights into the factors governing the relative strengths of these two signals.

(3) The authors employ lasers with varying pulse widths in their experiments. I understand that shorter pulses offer better time resolution but may come at the expense of energy resolution. Can the authors briefly discuss the potential influence of the limited energy resolution associated with the narrow pulses on the analysis of the experimental results?

(4) In Figure 3c, the temporal width of the FWM signal appears to exceed the duration of the probe pulse. Can the authors elaborate on the factors that might contribute to this broadening?

(5) The data presented in Figure 3f suggest that higher-energy hot carriers exhibit shorter lifetimes. While this trend seems plausible, I noticed some fluctuations in the measured curves. What is the origin of these fluctuations?

(6) Could the authors clarify the physical mechanism for the spatial distribution of the anti-Stokes and FWM processes observed in Figure 4? Do these processes predominantly occur in one of the electrodes (e.g., the tip), or are they primarily generated within the nanocavity formed by both the tip and the substrate?

Reviewer #3

(Remarks to the Author)

The manuscript by Luo et al. reports on an investigation of hot-carrier dynamics using atomic-scale nonlinear optical spectroscopy based on tip-enhanced near-field microscopy. The authors describe their experimental approach for probing non-equilibrium ultrafast carrier dynamics within plasmonic STM junctions, composed of an Au tip and an Au(111) surface. They analyze the time-dependent spectral response in two-color broadband spectroscopy, where inelastic light scattering (anti-Stokes) is observed. By carefully characterizing the spectral evolution and the excitation-fluence dependence, they interpret the inelastic light scattering spectra as a combination of four-wave mixing (FWM) and anti-Stokes electronic Raman scattering (ERS). Furthermore, they examine the tip-surface distance dependence of the nonlinear near-field response, demonstrating that the light-matter interaction is highly confined at the atomic-scale length and that localized hot-carrier dynamics can be resolved at this level. Their analysis also reveals the energy-dependent behavior of the inelastic light scattering spectra, distinguishing the distinct decay dynamics of FWM and ERS. In addition, to support their experimental observations, the authors perform simulations based on a finite element model combined with the three-temperature model and rate equations to account for local hot-carrier dynamics. Finally, they demonstrate that their technique can resolve the nonlinear optical response of a single graphene nanoribbon (GNR), underscoring its capability for characterizing atomic-scale inhomogeneity in hot-carrier dynamics.

As discussed in the introduction, hot-carrier dynamics in plasmonic nanostructures is a fundamental topic for understanding plasmon-induced phenomena. However, ultrafast timescales and spatial inhomogeneity have hindered a comprehensive understanding of these processes. The authors present a unique approach that enables the study of hot-carrier dynamics with both ultrafast time resolution and atomic-scale spatial resolution, providing new insights into its microscopic mechanisms. This method could be extended to other nanosystems, including single atoms and molecules. Given the broad significance of hot-carrier dynamics in plasmonic nanostructures and the potential for future advancements in tip-enhanced nonlinear near-field spectroscopy, this manuscript is expected to attract wide interest among researchers in nanoscale science and technology. Therefore, I recommend its publication in Nature Communications.

Before accepting the manuscript, I would appreciate it if the authors could address the following points:

1. In ACS Photonics 8, 2610 (2021), anti-Stokes scattering from plasmonic STM junctions was shown to originate from either electronic Raman scattering or photoluminescence, depending on the spectral overlap between the localized surface plasmon resonance and the excitation wavelength. Does the present study observe similar behavior?
2. The above reference also discusses the bias-voltage dependence of anti-Stokes scattering, which seems relevant to the description on page 4.
3. In ACS Photonics 10, 3637 (2023), the hot-carrier distribution was found to be reflected in the current-voltage (I-V) characteristics of the plasmonic STM junction under cw-laser excitation. It would be valuable to compare the I-V curve with and without laser excitation in the present study.
4. Ultrafast hot-electron dynamics in STM junctions has also been reported in ACS Nano 16, 14479 (2022). I believe this reference should be included.
5. Regarding the discussion of electronic Raman scattering, is the concept of "resonance" necessary? Since metals have a continuum of electronic states, a well-defined resonance behavior may not be clearly distinguishable. Could the authors clarify this point?
6. The authors should provide STS mapping data of the GNR to compare the spatial distribution of the local electronic DOS with the anti-Stokes mapping shown in Fig. 4. This would strengthen the interpretation of the observed spatial variations in nonlinear optical response. Ideally, the STS mapping data should be added in Fig. 4.
7. In Fig. 1c, it may be more appropriate to change the label from "LSP" to "STML" for clarity.
8. The authors mention "deep-sub-wavelength meta-atoms." Since the concept of meta-atoms may not be immediately clear, I recommend adding a reference, such as Nature Photonics 8, 889 (2014).
9. On page 3, the phrase "the edges of the GNR compared to its bulk" would be clearer as "the edges of the GNR compared to its interior."
10. On page 6, in P_{probe} and I_{FWM}, "probe" and "FWM" should not be italicized.

Version 1:

Reviewer comments:

Reviewer #1

(Remarks to the Author)

I am generally satisfied with the authors' responses to the previous round of reviews. They have addressed the key concerns and provided reasonable clarifications and additional context where needed.

While there are still some details in their explanations—particularly regarding the non-resonant plasmon excitation—that could be subject to debate, I do not believe it is necessary to pursue those points further in the context of this review. These are nuanced issues that are better left to ongoing discussion within the broader research community.

Overall, this manuscript presents a substantial and well-executed experimental and theoretical investigation. It represents a significant contribution to the ongoing debate on hot carrier generation in plasmonic systems and will be of interest to researchers working in the fields of nanophotonics and ultrafast dynamics. I recommend publication.

Reviewer #2

(Remarks to the Author)

The authors have properly addressed all my previous concerns. I therefore recommend the publication of this work in Nature Communications.

Reviewer #3

(Remarks to the Author)

The authors have satisfactorily addressed all of my comments by providing additional experimental data. I appreciate their effort and fully support the publication of this manuscript.

Response to the referees

We would like to cordially thank all the referees for their invaluable comments. We are happy to read that all the reviewers appreciated our manuscript, and judged it “an impressive combination of experimental and theoretical work, producing timely and potentially impactful results” (**reviewer #1**), “a significant advance in the field” (**reviewer #2**) and “a unique approach that enables the study of hot-carrier dynamics with both ultrafast time resolution and atomic-scale spatial resolution”, that “is expected to attract wide interest among researchers in nanoscale science and technology” (**reviewer #3**). We have revised the manuscript in accordance with their suggestions and criticisms, which we have fully addressed. We mark essential revisions as **REV#** and have underlined the new text (____) in the main text and supplementary materials (SM) to ease the evaluation.

We hope that the referees will acknowledge the significant improvements made in the manuscript, certainly driven by their keen comments, and will now be happy to recommend it for publication in Nature Communications. Below we enlist detailed responses to all comments of the referees.

REVIEWER COMMENTS

Reviewer #1 (Remarks to the Author):

The authors present a study on light upconversion in plasmonic structures and use it to explore the time-dependent properties of hot electrons excited in gold nanocavities. They use an STM tip coupled to a gold surface to achieve extreme localization of plasmonic fields in picocavities formed within the tunneling gap. Using pump-probe measurements, they characterize the nonequilibrium carriers on ultrafast timescales, shedding light on some of the most hotly debated issues in hot-electron physics such as the timescales of carrier relaxation and the nature of the electronic excitations in these systems. They also perform extensive modeling of the spatiotemporal dynamics of the excited carriers, accounting for various contributions, including both non-thermal and thermal carrier distributions. Finally, they apply this spatiotemporal mapping technique to graphene nanoribbons to image these properties at the atomic scale.

The paper presents an impressive combination of experimental and theoretical work, producing timely and potentially impactful results. However, several substantial issues in the data analysis currently prevent me from recommending the paper for publication.

We would like to sincerely thank the reviewer for his/her kind words and for judging our study “an impressive combination of experimental and theoretical work”, and our results “timely and potentially impactful”. We are glad to make all possible efforts to increase the clarity of all the points raised and to meet the expectations of the reviewer.

1. The plasmonic resonance of the system lies around 660–680 nm. This implies that, particularly in the pump–probe experiments, the excitation is non-resonant. Consequently, the discussion of hot-electron effects becomes questionable, because the processes described in Figure 2a appear to be purely plasmonic. Although there is likely some absorption at the pump and probe

wavelengths (as shown by Figure S6), it is doubtful that this absorption alone leads to any additional physics beyond what is observed in bulk gold.

We agree with the reviewer that the excitation pulse is non-resonant with our picocavity (as indicated by Fig. S9 in the revised SI), and detuned by ~ 170 nm from its resonant peak (~ 680 nm). As such, the ultimate mechanism of photoabsorption and hot carrier generation in the picocavity might not fundamentally differ from what occurs in bulk gold. In general, within our semi-classical approach, the non-resonant excitation does not alter the process of hot carrier generation. In fact, this is indeed why our model (although with a simplified formulation) was successfully applied to metal thin films in the past [Della Valle et al. *Phys. Rev. B* **86**, 155139 (2012)].

However, the critical difference between our system and, e.g., a non-resonant thin film lies in the extreme localisation of the incident electromagnetic fields of the laser pulses in the picocavity. By inspecting Fig. S9b (revised SI), it is clear that, although the pump wavelength is non-resonant, the absorption pattern at this wavelength is strongly affected by the cavity, and is reminiscent of the spatial distribution of the electric field near resonance. Notably, this is a distinctive feature of the picocavity.

To illustrate this effect, we have performed additional simulations of the picocavity for increasing distance between the Au substrate and the nanosphere mimicking the experimental STM Au nanotip. Figure R1 below shows the main results of our simulations for gap-sizes of 0.5 nm, 3 nm, 20 nm and 100 nm, from left to right, respectively. For each configuration, the absorption and scattering cross-sections are shown (top panels), together with the (normalised) absorption pattern at resonance (middle panels) and at a wavelength of 860 nm (bottom panels), which is the central wavelength of the pump pulse. Our numerical results suggest that, for the smallest gap, the dissipated electromagnetic power is reduced by a factor of ~ 20 for off-resonance excitation compared to the on-resonant excitation, yet it preserves a comparable spatial distribution. The field localisation across a broad spectral range is strong in the picocavity; hence, it governs the interaction with light even for wavelengths detuned from the resonant peak. However, even for a 3 nm gap, while the resonant mode is still confined in the cavity region, a non-resonant excitation exhibits a dissipation pattern that starts to spread over the surface of the substrate, thus producing a lesser degree of confinement. When the gap size increases further, the cavity mode is less and less defined (the nanosphere tends to behave as an isolated scatterer, with its absorption pattern), but, again, a non-resonant wavelength corresponds to an excitation delocalised over the metallic substrate, in sharp contrast to the spatial distribution for the picocavity with a gap-size of 0.5 nm.

Fig. R1 | Electromagnetic field localisation across the plasmonic picocavity. **a**, Simulated spectra of the absorption (left-axis) and scattering (right-axis) cross-sections of a picocavity with 0.5 nm gap-size. The same excitation parameters as in Fig. S9 were considered. **b**, Normalized spatial distribution (two-dimensional cross-cut) of the electromagnetic dissipated power $\rho_{\text{abs}}(\mathbf{r})$, evaluated at the picocavity scattering resonance, 690 nm. **c**, Same as **b** for a non-resonant wavelength of 860 nm. Note the multiplication factor in the spatial map, for the same colour bar as in **b**. **d-f**, Same as **a-c**, for a gap-size of 3 nm. **g-i**, Same as **a-c**, for a gap-size of 20 nm. **j-l**, Same as **a-c**, for a gap-size of 100 nm. The wavelengths (resonant and non-resonant) used in the spatial distribution maps of the electric field are annotated in the respective figures.

Following the reviewer's comments, we have now included Figure R1 in the SI (new Fig. S10, REV#1), alongside a paragraph to comment on the aspects mentioned above (see page #11 in SI). In the main text, on page #5 (REV#1), we have added the following sentence:

REV#1: 'Moreover, the extreme localization of the electromagnetic fields even far from its LSP (see Fig. S10) in the picocavity, ensures that both the pump and the probe pulses, although non-resonant, effectively interact with the system.'

2. The slope of 2 observed in Figure S2 can be explained by intraband PL rather than Raman scattering (see Nat Comm , 15, 4468 (2024) or ACS Nano 17, 11439-11453 (2023) for reference). A key question is whether the model presented in the Supporting Information can adequately fit the experimental results shown in Figure S2b.

Our model accurately captures the invariance of the spectrally resolved power-law exponent of 2, evaluated at various emission wavelengths of the anti-Stokes spectra and is consistent with the experimentally measured invariance of the power-law exponent. The revised version of Fig. S2b (reproduced below for convenience) compares the spectrally resolved power-law exponent as evaluated by our model and the experimentally measured one. Please see REV#2 in SI on page #3.

We have also included the reference 'Nat Commun. **15**, 4468 (2024)' as **Ref. #40** in the revised manuscript. The reference 'ACS Nano 17, 11439 (2023)' was already cited in our original submission, and it appears as **Ref. #36** in the revised main text.

Fig. R2 | Spectrally-resolved power-law exponent in the anti-Stokes spectra. The solid-blue curve represents the spectrally resolved power-law exponent extracted from the simulated anti-Stokes spectra.

3. In Figure 1e (green curve), does the simulated slope take into account the hot-electron Raman process and the rate-equation simulations described in the Supporting Information? Clarification on how these factors are incorporated into the modeling would be helpful.

In Fig. 1e, we had shown the simulated power-scaling of the anti-Stokes signal intensity for the single-pulse experiment. This simulation was performed using the same numerical approach and theoretical formalism detailed in the Supporting Information that was used to produce the anti-Stokes spectra in Fig. 3. Yet, unlike Fig. 3 where two laser pulses (a pump and a probe) interact simultaneously with the cavity, the calculations in Fig. 1e were conducted by (i) setting the power of the 'pump' pulse to zero; and (ii) considering only the 'probe' pulse interacting with the cavity. For each power of this single pulse simulation, the corresponding nonequilibrium electronic distribution $f(E,t)$ and joint density of electronic states $J_E(\omega_e,t)$ were calculated (following Eq. S6, S8, and S9), allowing us to define the anti-Stokes cross-section (Eq. S5). In other terms, compared to the simulations presented in Fig. 3, in Fig. 1e we had switched off the additional contribution arising from the pump pulse excitation in the carrier distribution $f(E,t)$, hence, in the energy distribution of the joint density of electronic states ρ_j , Eq. S6. The intensity of the anti-Stokes emission was taken at the zero delay with respect to the temporal peak of the pulse, and at a wavelength of 670 nm (consistent with the experimental conditions).

We have now included a statement in the SI to make it clearer, please see **REV#3** in SI on page #13. The edited sentence is also reproduced below for convenience:

[...measured experimentally]. **REV#3** *In particular, since no specific assumptions were made about the electron energy distribution $f(E)$ in deriving the equations above (Eqs. 5-7) to express σ_{aS} , the formalism outlined above was applied to describe both single-pulse experiments (Fig. 1 in the main text) and pump-probe measurements (Figs. 2 and 3 in the main text). ... “*

4. In Figure 1, the probe pulse alone is observed to produce anti-Stokes emission. However, in Figure 2, the presence of the pump pulse is required for anti-Stokes emission to be detected. What causes this discrepancy? Is it because of the low power of the probe pulse?

In our pump-probe experiments, the role of the pump pulse is to generate hot carriers in the picocavity, whereas the role of the probe pulse is to induce electronic resonance Raman scattering from these carriers, enabling their time-resolved detection via the anti-Stokes emission. The reviewer is correct that, in the spectral range of our measurement, we can measure the anti-Stokes signal generated by the probe pulse alone (as shown in Fig. 1). Since we intend to probe the dynamics of the hot carriers, the anti-Stokes signal generated by the probe pulse has been removed in the pump-probe spectra, by subtracting the spectrum measured at a negative delay of -1 ps, where the anti-Stokes signal solely arises from interaction with the probe pulse.

5. What can be inferred from the anti-Stokes scattering transient spectra regarding the carrier distribution—specifically, whether it follows a Fermi–Dirac distribution or a non-Fermi distribution similar to e.g. Nat. Comm., 9, 1853, (2018)? Can the relative population of non-Fermi electrons be estimated as a function of time (for example, from Figure S8c)?

According to our model, the transient anti-Stokes spectra (Fig. 3d) are dominated by the contribution arising from the nonthermal electrons, referred to as ‘non-Fermi electrons’ by the reviewer. The contribution of the thermal hot electrons, which populate nonequilibrium states in close proximity to the Fermi level, is essentially negligible in the simulated dynamics. To illustrate this clearly, we have disentangled our simulations of the pump-probe spectra by considering separately the contributions of nonthermal and thermal carriers. The two anti-Stokes emission maps, shown below in Fig. R3a and R3b, respectively, effectively show that the nonthermal contribution dominates over the thermal one by six orders of magnitude (note the multiplication factor in Fig. R3b).

Furthermore, from our simulations we can extract the temporal dynamics of nonthermal carriers (expressed by their excess energy content $N(t)$, according to the 3TM), and compare it directly with the evolution of the electronic temperature. These dynamics are shown in Fig. R3c and Fig. R3d, respectively. The corresponding variations of the electron energy occupancy distribution $\Delta f_{NT}(E,t)$, associated with nonthermal carriers, and $\Delta f_T(E,t)$, related to the thermalised hot electrons, are also shown in Fig. R3e and Fig. R3f, respectively.

We have now included Fig. R3 in the SI on page #22 (as new Fig. S14), alongside a full paragraph commenting each panel, please see **REV#4** in SI. The reference pointed out by the reviewer has also been included (it now appears as **Ref. #28** in SI), as it was very relevant to our discussion. In the main text, we have included (**REV#4** on page #8) an explicit reference to this new SI figure.

Fig. R3 | Contribution of the nonthermal and thermal carriers in the measured dynamics. **a**, Disentangled differential map of the ultrafast modulation of the anti-Stokes spectra as a function of the pump-probe time delay, due to nonthermal carriers only. The same excitation conditions as in Fig. 3 of the main text were considered in the simulation. **b**, Same as **a**, disentangling the contribution for the thermal carriers only. Both **a** and **b** are normalized to the maximum value in **a**, with a 10^6 multiplication factor in **b**. **c**, Ultrafast dynamics of $N(t)$, the excess energy density stored in the nonthermal fraction of the population of carriers photoexcited by the pump pulse. **d**, Ultrafast dynamics of the corresponding electronic temperature increase, $\Delta\Theta_e(t)$. **e**, Photoinduced variation of the electron energy occupancy distribution due to the nonthermal carriers, Δf_{NT} , at the annotated pump-probe delays. **f**, Same as **e** for the thermal carriers and the corresponding variation of Δf_T .

6. The decay constants shown in Figure S8 are presumably determined by the parameters in Equation S10 of the Supporting Information. How are these parameters (e.g. τ_0) chosen or derived from the data?

The reviewer is right, the decay constants were determined by the parameters in Equations 10 (or, equivalently, Equations 12 for the specific quantities shown in Fig. S8). Apart from $t_{diff,eff}$ (which is a fitting coefficient, extensively commented in SI section VI), all other parameters in Eqs. 10a-c and 12a-c were taken from previously reported literature (see e.g. ref. 17 of our originally submitted Supporting Material and references therein). Regarding τ_0 , we set its value to 6 fs in agreement with previous studies of ours, where we applied our model to analyse ultrafast pump-probe measurements, and successfully reproduced experimental results with quantitative accuracy. Although 6 fs is slightly larger compared to the value predicted by Fermi liquid theory (which is closer to 1 fs, see e.g. [Carpene, *Phys. Rev. B* **74**, 024301 (2006)]), this increase is in

fact consistent with more advanced, atomistic calculations. Indeed, as reported e.g. in [ACS Energy Lett. 4, 2552 (2019)], quantum kinetics calculations indicate that the nascent distribution of out-of-equilibrium electrons ought to comprise more prominent contributions near the Fermi level compared to the simpler assumption of a homogeneous energy distribution typically used in our 3TM. Electronic states closer to the Fermi level are associated with lower electron-electron scattering rates, which should correspond to an increased timescale for the effective internal thermalisation of carriers. De facto, we account for this effect by increasing the value of τ_0 .

Following the reviewer's comment, to clarify this point, we have edited the paragraph commenting on the coefficients in the 3TM as follows (see page #15 of the SI, **REV#5** in SI):

"...EF the Au Fermi energy, and τ_0 a material's constant, here increased to 6 fs relative to estimates from Fermi liquid theory to reflect the lower scattering rates predicted by quantum kinetics calculations, a modification that has been shown to accurately match ultrafast pump-probe experiments..."

7. Why does the graphene local density of states (LDOS) influence the electronic processes in gold, given the earlier assertion that anti-Stokes emission originates primarily from hot electrons in Au? Is there additional Raman scattering from carriers in graphene, or another mechanism at play?

A graphene nanoribbon (GNR) lying flat on the Au(111) surface constitutes a coupled electronic system, where the electronic levels of the GNR hybridize strongly with that of the Au(111) surface. This hybridization makes the photoexcitation of hot carriers occur within the coupled electronic system, rather than from a specific entity in the picocavity (nanotip, Au(111) surface or the GNR). Isolating the contribution of the GNR alone in the anti-Stokes emission would require placing the GNR on top of an ultrathin decoupling layer (e.g., few layers of NaCl) on top of Au(111) surface, which is beyond the scope of the current work. Nevertheless, we do not see clear Raman scattering from carriers in the graphene nanoribbon, as the characteristic energies (e.g. 1580 cm^{-1}) would correspond to emission wavelengths in the spectral range of $\sim 620\text{-}650 \text{ nm}$, which is considerably lower than the analyzed (and presented) hot carrier signal.

8. Can the lifetimes of electron thermalization as a function of electron energy (shown in Figure 3f) be assessed theoretically? They appear consistent with Fermi liquid theory, so it would be valuable to see theoretical insights on this behavior. Furthermore, the timescale of non-thermal electron decay is notably slow compared with previous literature—does any existing theory explain such slow decay?

(i) Regarding the energy-dependent lifetime of hot carriers: the model used in the current work, being an extension of the 3TM, does not incorporate an energy dependence for the thermalization dynamics of hot carriers. Indeed, this represents a foundational assumption of the 3TM. A variant of the 3TM would therefore be required for accessing the energy dependent relaxation times of the hot carriers. The extended Two-Temperature Model (E2TM) [Carpene, *Phys. Rev. B* **74**, 024301 (2006)] would be suitable for this purpose, yet at the expense of numerical complexity. Until date, the E2TM has been primarily limited to thin films of metals and homogeneous nanoparticles. An extension of the E2TM to more complex geometries (like our picocavity) would

require efforts beyond the scope of the current work. We thank the reviewer for the valuable remark, which we will certainly consider for future extensions of our modelling approaches.

(ii) Regarding the timescales of nonthermal electrons: according to the results published e.g. in Nature Commun. **6**, 7044 (2015), the total rate of electron scattering (including electron-electron and electron-phonon processes) corresponds to a time constant of ~30 fs. We assume that the reviewer refers to these as the typical timescales for nonthermal electron decay. However, this value should not be considered as the time constant of the full thermalization process of the electronic subsystem, as pointed out e.g. in ACS Photonics, **5**, 2584 (2018). Even though, the individual scattering events occur on a timescale of a few femtoseconds, the complete thermalization requires hundreds of these events, and the overall relaxation time of the nonthermal carriers is thus estimated to be on the order of several hundreds of femtoseconds, which is comparable with the decay timescales predicted by our model. In addition, please note that the quantum kinetics calculations reported by Govorov's group, see e.g. ACS Energy Lett., **4**, 2552 (2019), also suggest an increased timescale for internal electron thermalization. Overall, these reports from the literature corroborate our findings and the characteristic timescales derived from our model.

Considering the Reviewer's remarks, to further clarify these aspects, we have included the following statements in the SI (**REV#6** on pages #14 and #22 respectively):

(i) “[... Fermi-Dirac distribution given above]. **REV#6** *In passing, note that Eq. S8 treats the energy and temporal dependences as separable, in $\delta_{NT}(E)$ and $N(t)$, respectively, implying that all nonthermal carriers relax at the same rate (following electron-electron scattering events, see below for details). This assumption is foundational to the 3TM, and incorporating an energy-dependent carrier relaxation would require alternative modelling approach, with increased numerical complexity, hence falling beyond the scope of this work.*”

(ii) “[...few hundreds of femtoseconds]. **REV#6** *This timescale is consistent with prior experimental observations Ref. #19, #28 and #29 and theoretical descriptions of the scattering events experienced by photoexcited electrons towards equilibrium Ref. #21, #25, #30.*”

9. The results presented in Figure S8 suggest that, under this experimental configuration, the optical response is dominated by a nonthermal, non-Fermi distribution of electrons—an unusual observation compared to most literature reports. These findings should be highlighted more prominently in the main text.

We thank the reviewer for pointing this out. We have now included a statement in the introductory part of the main-text highlighting this point. Please see **REV#7** on page #3 of the main text. ‘*Our analysis reveals that the dominant contribution to the ultrafast modulation of anti-Stokes emission arises from photoexcited hot carriers exhibiting a nonthermal distribution, with energies significantly far from the Fermi level*’

10. Finally, does the tunneling current affect any of the optical or electronic phenomena reported in the paper? A brief discussion or control experiment would help clarify its impact.

In our experiments, since the STM was operated in the constant-current mode, the variation of the tunneling current leads to a change in the size of the picocavity. Thus, a higher tunneling

current implies a smaller vertical size of the picocavity compared to a lower tunneling current. The variation in the intensity of the anti-Stokes signal as a function of reducing tunneling current or increasing size of the picocavity is shown in Fig. 1d (main text). The intensity of the anti-Stokes signal decreases exponentially upon reducing the tunneling current in the cavity, primarily due to the changes in the local field enhancement within the cavity. The number of tunneling electrons, by contrast, has a negligible influence on the optical or electronic response in the measured spectra.

Reviewer #2 (Remarks to the Author):

By using nonlinear optical spectroscopy in a plasmonic cavity formed in the tunnel junction of a scanning tunneling microscope (STM), the manuscript by Luo et al. reports a novel method for visualizing hot carrier dynamics at the atomic scale. The authors convincingly demonstrate the ability to track hot carriers and map their spatial distribution, revealing energy-dependent relaxation times and the influence of local density of states. The combination of ultrafast pump-probe spectroscopy with STM offers a unique approach to accessing hot carrier dynamics, with the capability to achieve exceptionally high resolution in both temporal and spatial domains. The experimental results are well supported by theoretical calculations and simulations, providing a comprehensive understanding of the observed phenomena.

I believe that this work represents a significant advance in the field. Therefore, the paper can be considered for publication in Nature Communications if the authors can properly address the following comments or concerns:

We would like to kindly thank the reviewer for recognizing the novelty of our work and for considering the “exceptionally high resolution in both temporal and spatial domains” to be “a significant advance in the field”. We are glad to make all possible efforts to increase the clarity of all the points raised and to meet the expectations of the reviewer.

(1) In Figure 1c, the experimental LSP electroluminescence and anti-Stokes spectra show some fine structures, while the simulated LPDOS appears smoother. Can the authors comment on the possible origin of this difference?

We would like to thank the reviewer for raising this point. There are indeed fine structures in the LSP and the anti-Stokes spectra, as the reviewer points out. These fine structures are plausibly the consequence of atomic scale structural variations at the very end of the apex of the nanotip. These fine structures can change on atomic-scale modification (e.g., indentation) of the nanotip. A deeper understanding of this phenomenon would require extensive FDTD simulations in the quantum regime, which are beyond the scope of the current work. Nevertheless, we would like to stress that the measurements reported in the current work were performed with the identical conditions of the nanotip. We have now added to the main text a sentence explaining the origin of the observed fine structures in the electroluminescence and anti-Stokes spectra. Please see **REV#8** on page #4 in the main-text.

REV#8 *It is worth mentioning that fine structures present in the anti-stokes spectrum and the LSP in Fig. 1c plausibly originates from the atomic-scale structural variations at the very end of the apex of the nanotip, which were not considered in the simulations (green curve in Fig. 1c) ’*

(2) In the zero-delay experiment presented in Figure 2b, the four-wave mixing (FWM) and anti-Stokes signals exhibit comparable magnitudes despite their distinct physical origins. I wonder if the authors could provide some insights into the factors governing the relative strengths of these two signals.

We thank the reviewer for raising this interesting point. The FWM shown in Fig. 2d involves two photons from the probe pulse and one photon from the pump pulse. The anti-Stokes signal, on the other hand, originates from the scattering of the probe pulse by the hot carriers excited by the pump pulse. The relative intensities of these two contributions can be varied by tuning the plasmonic cavity (e.g. shaping the nanotip by indentation). The FWM signal can become comparable to the anti-Stokes signal as the probe pulse is in resonance with the LSP of the cavity as shown in Fig. 2d. Therefore, the interplay between the resonance condition and the cavity geometry is likely to play a crucial role in determining the relative intensities of the FWM and anti-Stokes signals in the experiment.

(3) The authors employ lasers with varying pulse widths in their experiments. I understand that shorter pulses offer better time resolution but may come at the expense of energy resolution. Can the authors briefly discuss the potential influence of the limited energy resolution associated with the narrow pulses on the analysis of the experimental results?

Pump and probe laser pulses with broader spectral bandwidth provide higher time resolution, as the reviewer correctly states. However, they also make the FWM and the hot carrier contributions commensurately broad, making it difficult to separate their contribution in the measured anti-Stokes spectra. The spectral separation and the duration of the pump and probe laser pulses were optimized in the experiments to make this separation feasible. We would like to note that the duration of the used laser pulses provides sufficient time resolution (~ 30 fs) to probe the hot carrier dynamics, which evolves on the time scale of 100 – 200 fs (Fig. 3f, main-text). At the same time, it provides sufficient energy resolution (~ 65 meV) to resolve the broadband hot carrier dynamics.

(4) In Figure 3c, the temporal width of the FWM signal appears to exceed the duration of the probe pulse. Can the authors elaborate on the factors that might contribute to this broadening?

The FWM signal overlays on the broad anti-Stokes signal generated by the hot carriers. In the spectral region of the FWM, therefore, there is an underlying contribution of the hot carriers, which is very hard to separate, even by varying the spectral separation between the pump and probe laser pulses. Since FWM is a relatively narrow spectral feature (defined by the bandwidth of the used laser pulses), it can still be discerned, nonetheless, it cannot be totally isolated from the contribution of the hot carriers.

To ensure that the temporally broadened FWM response observed in the anti-Stokes spectra is not resulting from the uncompensated chirp of the laser pulses, we have systematically measured the pump-probe spectra as a function of the varying chirp of the laser pulses (please see Fig. S7 in SI).

(5) The data presented in Figure 3f suggest that higher-energy hot carriers exhibit shorter lifetimes. While this trend seems plausible, I noticed some fluctuations in the measured curves. What is the origin of these fluctuations?

Figure 3f indeed show that the relaxation times of the hot carriers are energy dependent – high-energy hot carriers relax faster compared to the low-energy hot carriers. The fluctuating behaviour

in the relaxation time is likely an experimental limitation (e.g., randomly generated noise in the spectra), as it lies mostly within the error bar.

(6) Could the authors clarify the physical mechanism for the spatial distribution of the anti-Stokes and FWM processes observed in Figure 4? Do these processes predominantly occur in one of the electrodes (e.g., the tip), or are they primarily generated within the nanocavity formed by both the tip and the substrate?

We performed simulations to estimate the change in the local photonic density of states (LPDOS) in the picocavity when a graphene nanoribbon (GNR) is present in the sub-nm gap between the STM tip and the flat Au(111) surface, as described in **section VIII of the SI**. The mere change of the tip height while scanning the GNR in the constant current mode of the STM does not yield the behavior observed in the experiments, i.e., enhanced FWM and hot carrier signals at the edges of the GNR. In the simulations, only the dielectric properties of the GNR were considered, not its intrinsic electronic structure. This simulation corroborates the fact that the observed features were not a geometrical artefact.

The differential conductance mapping of the GNR (also suggested by reviewer #3), measured in absence of the laser pulses, now presented in Fig. S4 shows the presence of the edge states of the GNR. Please see **REV#9** in the SI (page #4) and the main text (page #9). A line-profile of the differential conductance was shown in the previous version of the manuscript (Fig. S3). Please note that this GNR is not the same one as used in Fig. 4 (main-text), nevertheless, it does not alter any conclusions.

We have reproduced the Fig. S4 as Fig. R5 below for the convenience of the reviewer.

The simulations and the differential conductance measurements strongly indicate that the FWM and the hot carrier signals are enhanced due to the edge states of the GNR. A higher electronic density (DOS) would make available a larger pool of carriers which can be photoexcited by the incident pump pulses, leading to a stronger anti-Stokes signal from the hot carriers at the edges of the GNR compared to its bulk (Fig. 4b). In close analogy to the hot carrier signal in the anti-Stokes spectra, the higher DOS at the edges of the GNR could enhance the nonlinear susceptibility to the incident light. As a result, the nonlinear response linked to the third-order nonlinear susceptibility ($\chi^{(3)}$) exhibits atomic-scale variations within the GNR.

Fig. R5 | Differential conductance mapping of a GNR. **a**, Constant current STM topographic image of a GNR. The colour bar indicates the variation of the vertical position of the nanotip. **b**, Simultaneously recorded differential conductance map of the GNR at the bias of 1 V. A set point current of 100 pA and a bias of 1 V was used for the STM and STS mapping. The colour bar denotes the value of the differential conductance in pS.

Reviewer #3 (Remarks to the Author):

The manuscript by Luo et al. reports on an investigation of hot-carrier dynamics using atomic-scale nonlinear optical spectroscopy based on tip-enhanced near-field microscopy. The authors describe their experimental approach for probing non-equilibrium ultrafast carrier dynamics within plasmonic STM junctions, composed of an Au tip and an Au(111) surface. They analyze the time-dependent spectral response in two-color broadband spectroscopy, where inelastic light scattering (anti-Stokes) is observed. By carefully characterizing the spectral evolution and the excitation-fluence dependence, they interpret the inelastic light scattering spectra as a combination of four-wave mixing (FWM) and anti-Stokes electronic Raman scattering (ERS). Furthermore, they examine the tip-surface distance dependence of the nonlinear near-field response, demonstrating that the light-matter interaction is highly confined at the atomic-scale length and that localized hot-carrier dynamics can be resolved at this level. Their analysis also reveals the energy-dependent behavior of the inelastic light scattering spectra, distinguishing the distinct decay dynamics of FWM and ERS. In addition, to support their experimental observations, the authors perform simulations based on a finite element model combined with the three-temperature model and rate equations to account for local hot-carrier dynamics. Finally, they demonstrate that their technique can resolve the nonlinear optical response of a single graphene nanoribbon (GNR), underscoring its capability for characterizing atomic-scale inhomogeneity in hot-carrier dynamics.

As discussed in the introduction, hot-carrier dynamics in plasmonic nanostructures is a fundamental topic for understanding plasmon-induced phenomena. However, ultrafast timescales and spatial inhomogeneity have hindered a comprehensive understanding of these processes. The authors present a unique approach that enables the study of hot-carrier dynamics with both ultrafast time resolution and atomic-scale spatial resolution, providing new insights into its microscopic mechanisms. This method could be extended to other nanosystems, including single atoms and molecules. Given the broad significance of hot-carrier dynamics in plasmonic nanostructures and the potential for future advancements in tip-enhanced nonlinear near-field spectroscopy, this manuscript is expected to attract wide interest among researchers in nanoscale science and technology. Therefore, I recommend its publication in Nature Communications.

We would like to sincerely thank the reviewer for his/her kind words and for recognizing a series of novel elements in our work, which are “expected to attract wide interest among researchers in nanoscale science and technology”. We have made all possible efforts to increase the clarity of the presentation in order to address all the points raised and to meet the expectations of the reviewer.

Before accepting the manuscript, I would appreciate it if the authors could address the following points:

1. In ACS Photonics 8, 2610 (2021), anti-Stokes scattering from plasmonic STM junctions was shown to originate from either electronic Raman scattering or photoluminescence, depending on the spectral overlap between the localized surface plasmon resonance and the excitation wavelength. Does the present study observe similar behavior?

Thank you for mentioning this point. In our study, we used ultrashort laser pulses with considerable spectral bandwidth, a continuous variation of the central wavelength of the laser pulses to investigate the nature of the anti-Stokes signal when the spectrum of the laser pulses is resonant or non-resonant with the local plasmonic response of the picocavity is very difficult. Nonetheless, the power-scaling experiments (Fig. 1e, main-text) ascertain that the electronic resonance Raman scattering of the hot carriers is the dominant mechanism behind anti-Stokes signal generation in our study.

2. The above reference also discusses the bias-voltage dependence of anti-Stokes scattering, which seems relevant to the description on page 4.

Figure R6 below shows a series of anti-Stokes spectra, measured at the zero time delay between the pump and the probe pulses for various biases applied to the picocavity. The cut-off energy of the anti-Stokes signal is independent of the applied bias in the STM junction. We have now included the following figure in the SI; please see **REV#10** on page#5 of the SI. This measurement further substantiates the electronic resonance Raman scattering mechanism.

Fig. R6 | Bias dependence of the anti-Stokes spectra. A series of anti-Stokes spectra, measured at the zero time delay between the pump and the probe pulses, for various biases applied to the STM junction. The biases used are annotated to each anti-Stokes spectrum. The spectra are vertically shifted for clarity. The STM junction was operated in the constant current mode with the set tunneling current of 1 nA.

3. In ACS Photonics 10, 3637 (2023), the hot-carrier distribution was found to be reflected in the current-voltage (I-V) characteristics of the plasmonic STM junction under cw-laser excitation. It would be valuable to compare the I-V curve with and without laser excitation in the present study.

We thank the reviewer for this comment. Figure R7 below shows the I-V measurement with and without illumination of the laser pulses. The I-V characteristics of the picocavity is modified with and without illumination by the laser pulses, as the reviewer correctly points out. Substantial laser-induced tunneling current is measured at the zero bias in the STM junction (red curve). The laser-induced tunneling current in this case is a combination of the photocurrent and the contribution of

the hot carriers. We have now included this figure in the revised SI, please see **REV#11** on page #5 of the SI.

Fig. R7 | Comparison of the variation of the I-V characteristics as a function of the applied bias in the STM junction, in presence (red curve) and absence (blue curve) of illumination with the laser pulses. Substantial laser-induced tunneling current is measured as visible from a non-negligible current (~ 7 pA) at the zero bias. Tunneling current of 100 pA at the bias of 1V was used for both I-V measurements. A single laser pulse of ~ 8 mW power as described in Fig. 1 (main text) was used for this measurement.

4. Ultrafast hot-electron dynamics in STM junctions has also been reported in ACS Nano 16, 14479 (2022). I believe this reference should be included.

We have now included the above reference in the revised text, please see **Ref. #24**.

5. Regarding the discussion of electronic Raman scattering, is the concept of “resonance” necessary? Since metals have a continuum of electronic states, a well-defined resonance behavior may not be clearly distinguishable. Could the authors clarify this point?

We agree with the reviewer that the metals have a continuum of electronic states. In metals, **the electronic Raman scattering process occurs between real electronic states and not virtual states**. There is no fundamental difference from the well-defined two-energy-level resonance case, e.g. in a single molecule. The convention used extensively in the literature to describe this mechanism of the anti-Stokes signal generation is ‘*electronic resonance Raman scattering*’, see for instance this review article: ACS Nano 15, 5785-5792 (2021). We have cited this reference for using this conceptual definition in the main text.

6. The authors should provide STS mapping data of the GNR to compare the spatial distribution of the local electronic DOS with the anti-Stokes mapping shown in Fig. 4. This would strengthen

the interpretation of the observed spatial variations in nonlinear optical response. Ideally, the STS mapping data should be added in Fig. 4.

We sincerely thank the reviewer for this suggestion. We have now included the STS mapping of a GNR in the SI. Please see **REV#9** and **Fig. S4** in the revised SI (page #4) and the main text (page #9). We have reproduced below the figure for convenience as Fig. R6. A line-profile of the differential conductance was shown in the previous version of the manuscript (Fig. S3). Please note that this GNR is not the same one as used in Fig. 4 (main-text), nevertheless, it does not alter any conclusion. Since the GNR used in the STS mapping and the one used in Fig. 4 (main-text) are not the same, in our opinion, it is better not to put them together in a Figure.

Fig. R5 | Differential conductance mapping of a GNR. **a**, Constant current STM topographic image of a GNR. The colour bar indicates the variation of the vertical position of the nanotip. **b**, Simultaneously recorded differential conductance map of the GNR at the bias of 1 V. A set point current of 100 pA and a bias of 1 V was used for the STM and STS mapping. The colour bar denotes the value of the differential conductance in pS.

7. In Fig. 1c, it may be more appropriate to change the label from "LSP" to "STML" for clarity.

We respectfully disagree with the reviewer on this suggested correction. The electroluminescence in the STM junction (referred to as STML) indeed provides information about the local surface plasmon (LSP) response. However, in our opinion, the LSP is the correct terminology to describe the plasmon response; we have explicitly mentioned in the text that it was measured via electroluminescence.

8. The authors mention "deep-sub-wavelength meta-atoms." Since the concept of meta-atoms may not be immediately clear, I recommend adding a reference, such as Nature Photonics 8, 889 (2014).

Thank you for this suggestion. We have now included the above reference in the revised text. Please see **Ref. #56**.

9. On page 3, the phrase “the edges of the GNR compared to its bulk” would be clearer as “the edges of the GNR compared to its interior.”

Thank you for mentioning this, this statement is indeed clearer. We have now revised it; please see **REV#12** in the main text (page #3).

10. On page 6, in P_probe and I_FWM, "probe" and "FWM" should not be italicized.

Thank you for pointing this out, we have now corrected it in the revised text.